# SELU: TARGETTED LLM UNLEARNING USING ENERGY

## ABSTRACT

Large language models (LLMs) often memorize sensitive or copyrighted content, motivating *machine unlearning* methods that can remove specific knowledge without retraining from scratch. A challenge arises from how fine-tuning is performed, where it uses a lower learning rate than pre-training to avoid destabilizing existing knowledge which leave models underconfident on data it wants to retain. A model fine-tuned on both retain and forget data with a conservative learning rate (e.g., 1e-5) differs from a retain-only model trained more aggressively (e.g., 1e-4), which achieves stronger likelihood-scale alignment which results in lower negative log-likelihood (NLL) on retained knowledge. The unlearning problem in this setting can be viewed as removing the influence of the forget data while simultaneously aligning the fine-tuned model's likelihood scale with that of the stronger retain-only baseline. We propose *Straight-through Energy Language Unlearning* (SELU), a parameter-efficient framework that integrates Low-Rank Adaptation (LoRA) with an energy-based objective guided by straight-through estimators (STE). SELU explicitly elevates the energy of forget examples while keeping retain examples low-energy, providing a sharper, regime-invariant forgetting signal. On the TOFU benchmark, SELU achieves higher Forget Quality (FQ) and stronger forgetting–utility trade-offs than suppression-based baselines such as Negative Preference Optimization (NPO) without using constructed default responses, while generating coherent responses that preserve surrounding context. Ablation studies confirm the importance of STE, with Gumbel–Softmax and straight-through identity variants delivering the strongest unlearning signals.

## 1 INTRODUCTION

Large language models (LLMs) have achieved remarkable success across many natural language tasks, yet their reliance on large-scale internet corpora has raised concerns such as memorising sensitive, copyrighted, or potentially harmful content (Carlini et al., 2021; Eldan & Russinovich, 2023). This creates risks of privacy breaches, copyright infringement, and unsafe generations. Legal frameworks such as the GDPR (Mantelero, 2013) and CCPA (Pardau, 2018) further underscore the importance of enabling the *right to be forgotten*, which necessitates mechanisms to remove specific information from trained models. In this context, the field of *machine unlearning* (Cao & Yang, 2015; Bourtoule et al., 2021; Yao et al., 2024) emerges with the goal to selectively remove knowledge without degrading model utility.

A naive solution to machine unlearning would be to retrain from scratch on a filtered dataset, known as *exact unlearning*, but automated data filtering is difficult and full retraining of LLMs is expensive. This motivate research on practical methods for *approximate unlearning* or *inexact unlearning*. Recent work has explored optimization-based approaches such as Gradient Ascent (GA) (Jang et al., 2023b; Yao et al., 2024) and preference objectives like negative-preference optimization (NPO) (Zhang et al., 2024). Yet, these methods often degrade text coherence (as in GA) or enforce rigid fallback responses (e.g., "I don't know") across all undesirable prompts, which limits to target the removal of specific facts or entities while preserving the surrounding linguistic structure.

A further challenge is that fine-tuning is usually carried out with a lower learning rate than pre-training, in order to make small, stable adjustments without destabilizing existing knowledge. While this conservative learning rate prevents catastrophic forgetting (Kirkpatrick et al., 2016), it also

leaves the model under-confident in its newly acquired retain knowledge. For unlearning, however, the ideal target is not just a model that forgets, but one that is also confident on the retain set meaning that it approaches the likelihood calibration of stronger retain-only baseline. This creates a gap between the conservatively fine-tuned forget+retain model and the stronger retain-only baseline, motivating the need for an unlearning objective that can bridge this calibration gap while removing the influence of the forget data.

In this paper, we introduce SELU, a parameter-efficient framework that addresses these shortcomings by combining low-rank adaptation (LoRA) with an energy-based objective. SELU leverages straight-through estimators to project discrete answer tokens into a differentiable energy function, enabling targeted forgetting of specific knowledge spans. By assigning high energy to forget examples (examples to remove) and low energy to retain examples (examples whose influence to keep), SELU provides a sharper and more localized forgetting signal compared to suppression-based techniques while aligning the model with the stronger calibration of the retain-only regime, yielding more confident retention alongside reliable forgetting even under learning rate mismatches.

**Contribution** This paper makes the following contributions:

1. We introduce SELU, a novel energy-guided framework for LLM unlearning that leverages straight-through estimators to directly elevate the energy of unwanted answer tokens for targeted forgetting without relying on constructed reference responses.

2. We conduct experiments on the TOFU benchmark with LLaMA-2-7B, showing that SELU achieves stronger forgetting–utility trade-offs than suppression-based baselines, but also provides reliable unlearning under calibration gaps caused by *learning rate mismatch*. SELU matches adjacent methods at $\approx 0.3$ for model utility while achieving substantially higher forget quality of $\approx 0.3$.

3. We provide ablation studies analyzing the role of straight-through estimators, highlighting the importance of Gumbel–Softmax in shaping effective unlearning signals. Our ablation shows that Gumbel-Softmax achieves the best forget quality ($\approx 0.3$) whilst having similar model utility ($\approx 0.3$) to other well performing estimators.

## 2 RELATED WORK

**Machine Unlearning for LLMs** Machine Unlearning (MU) has become increasingly important for LLMs, which often memorize sensitive, copyrighted, or harmful content from training corpora. Unlike typical classification tasks, unlearning for LLMs has to balance forgetting specific knowledge (the target) with maintaining the general fluency and reasoning ability (the constraint). Current approaches in MU are largely optimization-based, where models are fine-tuned on forget sets to reduce the likelihood of unwanted outputs. Gradient Ascent (GA) (Graves et al., 2021; Jang et al., 2023a; Yao et al., 2024; Maini et al., 2024) is a common baseline, but is unstable and prone to catastrophic forgetting and generating incoherent responses. More recently, preference-based objectives such as DPO (Rafailov et al., 2024) and NPO (Zhang et al., 2024) recast unlearning as an alignment problem to leverage toolkit from human preference tuning of LLMs. Other work explores lightweight interventions that includes unlearning layers (Chen & Yang, 2023), task vectors (Ilharco et al., 2023), and in-context unlearning using prompt engineering (Pawelczyk et al., 2024; Thaker et al., 2024).

**Parameter-Efficient Fine-tuning** Fine-tuning large language models is expensive due to their scale that requires billions of parameters to be updated. Parameter-Efficient Fine-Tuning (PEFT) offers an alternative that updates only a small subset of parameters. This approach allows the models to adapt to new tasks with significantly reduced compute compared to standard fine-tuning. Early PEFT approaches involved updating selected parts of the model such as the embedding layer (Artetxe et al., 2020; Ansell et al., 2022; Marchisio et al., 2023; Chen et al., 2023; Zhao et al., 2024), bias terms (Ben Zaken et al., 2022), normalization layers (Basu et al., 2023), or other transformer components. Later work introduced additional task-specific modules such as adapters (Houlsby et al., 2019; Pfeiffer et al., 2021), prefix-tuning (Li & Liang, 2021), and further prompt-based methods that seeks to augment the model with lightweight trainable components while leaving the base model intact. Low-Rank Adaptation (LoRA) (Hu et al., 2021) has recently become one of the most popular parameter-efficient fine-tuning methods due to its scalability and the effectiveness of its low-rank factorization structure. While LoRA applies this idea to neural network adaptation, low-rank factor-

ization itself has long been a core technique in areas such as recommender systems and knowledge graph completion (Koren et al., 2009; Lacroix et al., 2018; Chen et al., 2022). After fine-tuning, the trainable low-rank matrices can be merged into the frozen base model weights, ensuring no inference overhead. Numerous extensions have emerged after the inception of LoRA (Meng et al., 2025; Zhang et al., 2023).

# 3 PRELIMINARIES

**Problem Formulation for LLM Unlearning** The objective of machine unlearning for LLMs is to selectively remove the influence of a subset of training data, while preserving general model utility. Formally, let the *forget set* $\mathcal{D}_f = \{(x_i, y_i)\}_{i=1}^{N}$ denote examples that should be removed, and the *retain set* $\mathcal{D}_r = \{(x_j, y_j)\}_{j=1}^{M}$ denote examples whose performance should remain unaffected. The unlearning task is usually expressed as a regularized optimization problem balancing a forget loss $\ell_f$ with a retain loss $\ell_r$:

$$\min_{\theta'} \ \mathbb{E}_{(x,y)\in\mathcal{D}_f}[\ell_f(y \mid x; \theta')] + \lambda \, \mathbb{E}_{(x,y)\in\mathcal{D}_r}[\ell_r(y \mid x; \theta')]$$

where $\theta'$ are the parameters after unlearning, and $\lambda \geq 0$ controls the trade-off between forgetting and retention. In practice, the retain loss $\ell_r$ is often the cross-entropy loss, steering the model to maintain predictive performance on the retain set. The forget loss $\ell_f$ is defined to **discourage** the generation of the forget set, often by *inverting* or *perturbing* the prediction objective. While this framework provides a foundation for unlearning, its effectiveness depends heavily on the design of the forget loss and its interaction with the retain term.

**Energy-Based Models (EBM)** Energy-Based Models (EBMs) (Hinton, 2002; Lecun et al., 2006; Ranzato et al., 2007) provide a framework where an energy function $E_\theta(x, y)$ assigns a scalar score to each input–output pair $(x, y)$. Lower energy values correspond to more plausible outputs, while higher energies indicate implausible ones. Unlike likelihood-based modeling, EBMs require no explicit normalization, but instead shape the energy landscape such that desirable configurations occupy low-energy regions. In practice, EBMs for discrete data, such as texts, are often combined with pre-trained language models, which act as proposal distributions. This formulation, sometimes referred to as exponential tilting (Deng et al., 2021), defines a distribution as follows

$$p_\theta(x) \propto q(x) \exp\big(U_\theta(x)\big)$$

where $x$ is the text input, $q(x)$ is the base LLM, and $U_\theta$ is an auxiliary energy function. In this setting, the LLM *generates* candidate outputs, while the energy function *reshapes* their relative likelihoods so that the final output distribution can respond to suppress or encourage specific generations without requiring full retraining. Recent research highlight two integration modes where the LLM remains the generator while the energy function provides a supervisory signal that can be either fixed or adaptive. In the *static* mode, the energy model serves as an auxiliary loss or a filter during training, after which only the LLM is used at inference time (Lee et al., 2022). In the *dynamic* mode, the energy and language model are co-trained, with the energy model continually adapting to the evolving proposal distribution (Yoo & Lee, 2024).

**Straight-Through Estimators (STE)** A central challenge when applying EBMs to language generation lies in the discrete nature of texts. Unlike standard supervised training, where token targets are known and the model can be trained using a softmax layer and cross-entropy loss such as negative log-likelihood (NLL), EBM-guided learning applies a scalar energy score $U_\theta(x, y)$ to prompt-answer pairs $(x, y)$ and aims to propagate this signal back into the language model $q_\theta$. However, these answer sequences consist of discrete tokens selected from a finite vocabulary, which blocks gradient flow from the energy function back to the underlying logits. This challenge appears in two settings: (i) when the LLM first samples responses $\hat{y}$ to be scored by the energy function $U_\theta(x, \hat{y})$, and (ii) when the gold answer $y$ from the forget set is used directly. In both cases, the non-differentiable mapping from logits $z$ to discrete tokens prevents end-to-end training.

Straight-through estimators (STE) (Bengio et al., 2013) provide a practical solution to this problem by enabling gradients to flow through discrete sampling operations. The key idea is to use *hard* one-hot selections in the forward pass which ensures that the energy model evaluates realistic token sequences while using the corresponding *soft* distribution in the backward pass to approximate gradients. Variants include the straight-through logits (STL) estimator, which applies an

`argmax` followed by identity-gradient approximation, and the Gumbel-Softmax estimator (Jang et al., 2017), which introduces stochasticity via Gumbel noise to encourage exploration. Concretely given token-level logits $z \in \mathbb{R}^V$, STEs produce a hard one-hot vector in the forward pass $y_{\text{hard}} = \text{onehot}(\arg\max_i z_i)$ while in the backward pass they substitute its gradient with that of the soft probabilities $y_{\text{soft}} = \text{softmax}(z)$, $\frac{\partial \mathcal{L}}{\partial z} \approx \frac{\partial \mathcal{L}}{\partial y_{\text{soft}}}$ This can be written as the straight-through estimator:

$$\tilde{y} = (y_{\text{hard}} - y_{\text{soft}})_{\text{stopgrad}} + y_{\text{soft}} \qquad y_{\text{hard}} = \text{onehot}\left(\arg\max_i y_{\text{soft},i}\right)$$

which evaluates as $y_{\text{hard}}$ in the forward pass while backpropagating through $y_{\text{soft}}$.

The **STL** estimator uses the same pattern, but applies the straight-through operation directly on logits rather than softmax probabilities, effectively treating the $\arg\max$ as having an identity Jacobian during backpropagation

The **Gumbel-Softmax** estimator (Jang et al., 2017) introduces stochasticity to improve exploration, perturbing logits with Gumbel noise $g$ and temperature $\tau$:

$$y_{\text{soft}} = \text{softmax}\left(\frac{z + g}{\tau}\right)$$

**Low-Rank Adapation (LoRA)** Studies suggest that updates to large language models frequently lie in a low-dimensional subspace (Li et al., 2018). Instead of directly updating the full weight matrix $W \in \mathbb{R}^{d \times k}$ of a linear layer, LoRA parameterises the change $\Delta W$ as the product of two low-rank matrices, $A \in \mathbb{R}^{r \times k}$ and $B \in \mathbb{R}^{d \times r}$, where $r \ll \min(d, k)$ is the rank of the adapter. The output of the adapted layer for an input $x$ is then $(W + \Delta W)x = Wx + BAx$. During fine-tuning, the pretrained weight $W$ is kept fixed, and only the low-rank factors $A$ and $B$ are updated. This significantly reduces the number of trainable parameters while preserving the expressive power needed for adaptation. To avoid perturbing the model output at initialisation, $A$ is initialised with a small random distribution (He et al., 2015) and $B$ is initialised to the zero matrix. After training, the LoRA adapters can be merged back into the original weights, $W' = W + BA$, which ensures that inference speed and memory cost remain unchanged.

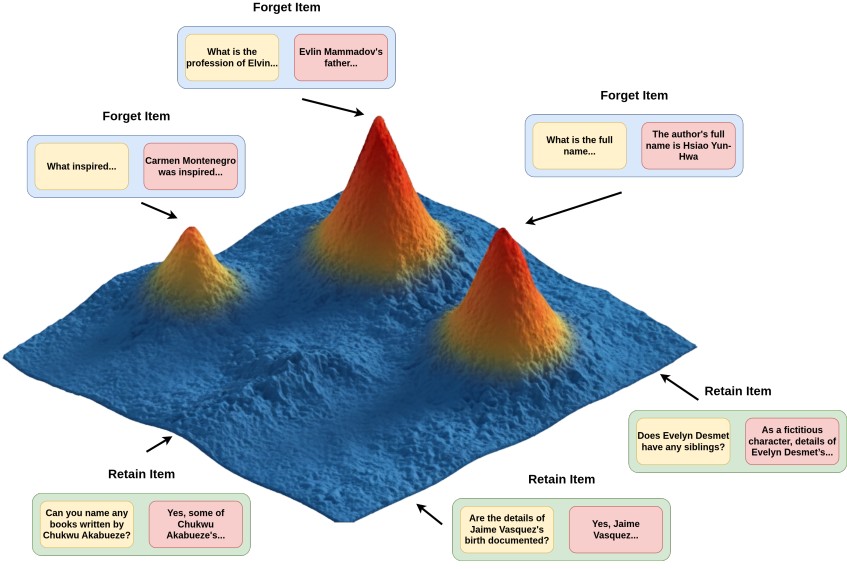

Figure 1: Loss Landscape with Energy $\mathcal{L}(E_\theta; \mathcal{D}_r, \mathcal{D}_f)$

## 4 STRAIGHT-THROUGH ENERGY FOR LLM UNLEARNING

We propose **Straight-through Energy for LLM Unlearning (SELU)**, a novel energy-guided objective that directly encodes the unlearning requirement into the model's optimization. SELU defines

a scalar energy function over prompt–answer pairs, trained to assign *low* energy to retain examples, to encourage confident retention, and *high* energy to forget examples, punishing over-confidence in information that must be removed as illustrated in Figure 1. Unlike preference-based approaches that indirectly discourage certain answers, SELU enforces an explicit energy gap between retained and forgotten knowledge, yielding more surgical forgetting while preserving general utility.

**Architecture** At the core of SELU is a lightweight two-layer MLP that maps structured prompt–answer representations into a scalar energy score. The prompt representation is pooled from masked hidden states ($h_{\mathrm{prm}}$), and the answer representation is obtained from STE-selected tokens projected into the embedding space ($h_{\mathrm{ans}}$). These are projected via $q = W_q h_{\mathrm{prm}}$, $k = W_k h_{\mathrm{ans}}$ and combined into a feature vector $\mathbf{f}(x, y) = [q,\, k,\, |q - k|,\, |\mathcal{P}|,\, |\mathcal{A}|,\, \cos(q, k)]$ which the MLP maps to the scalar energy $E_\theta(x, y)$. To enable differentiable selection of discrete tokens, SELU employs the Gumbel-Softmax straight-through estimator (Jang et al., 2017), ensuring that the energy function operates on one-hot token sequences in the forward pass while maintaining gradient flow in the backward pass. The full unlearning architecture can be found in Appendix A.

**Loss Design** The SELU loss integrates multiple terms to jointly optimize retention and forgetting so the model is confident in its retained knowledge while equally confident in not generating the forgotten data: $\mathcal{L} = \underbrace{\lambda_{\mathrm{CrossEntropy}} \mathrm{CrossEntropy}(y_r)}_{\ell_r} + \underbrace{\lambda_e \left(L_{\mathrm{down}} + L_{\mathrm{up}} + L_{\mathrm{pair}}\right)}_{\ell_r + \ell_f} + \underbrace{\lambda_{\mathrm{calib}} L_{\mathrm{cal}}}_{\ell_r} + \underbrace{\lambda_{\mathrm{cpl}} L_{\mathrm{cpl}}}_{\ell_r}$

The terms include (i) *unary push-down/up losses* which separately minimize retain energy $E_r$ and maximize forgetting energy $E_f$. $L_{\mathrm{down}} = \max(0,\, E_r - \tau_{\mathrm{low}})$; $L_{\mathrm{up}} = \max(0,\, \tau_{\mathrm{high}} - E_f)$ where $E_r$ and $E_f$ denote energies on retain and forget examples, $\tau_{\mathrm{low}}$ and $\tau_{\mathrm{high}}$ are thresholds, (ii) a *pairwise margin loss* that directly widen the gap between forget and retain examples, $L_{\mathrm{pair}} = \max(0,\, \mathrm{margin} + E_r - E_f)$ (iii) a *token calibration loss*, which refers to how well a model's confidence is in its output and which follows from Yoo & Lee (2024), aligning retain energies with token-level negative log-likelihood $L_{\mathrm{cal}} = \mathrm{MSE}(E_r, \alpha_{\mathrm{calib}} \cdot \mathrm{NLL}(y_r))$ (iv) a *token coupling loss* tying forget energies to gold-token probabilities $L_{\mathrm{cpl}} = \left(\sigma(E_f) - \bar{p}_{\mathrm{gold}}(y_f)\right)^2$, where $\bar{p}_{\mathrm{gold}}(y_f)$ is the average model probability assigned to gold tokens in the forget set. This formulation shapes the energy landscape so retain examples are pulled toward low energy, forget examples are pushed to high energy, and the pairwise margin ensures this separation. Together, these loss components stabilize optimization, via calibration and coupling that ties energy scores to token-level likelihoods, and prevent degenerated solutions such as flat energy functions or uniform suppression.

*Illustrative Example.* Consider the forget-set question:

> **Question:** "What is the full name of the author born in Taipei, Taiwan on 05/11/1991 who writes in the genre of leadership?"
> **Golden Answer:** "The author's full name is Hsiao Yun-Hwa."

In SELU, the straight-through estimator (STE) is applied to the *entire answer span* $\mathcal{A}$ as defined by the label mask (all positions with labels $\neq -100$). For each token $t \in \mathcal{A}$, the model produces logits $z_t \in \mathbb{R}^V$, which after Gumbel perturbation yield a hard one-hot vector in the forward pass:

$$\hat{y}_t = \mathrm{onehot}\left(\arg\max \mathrm{softmax}((z_t + g)/\tau)\right)$$

This operation is repeated for every token in the span $S = \{\text{"The"}, \text{"author's"}, \ldots, \text{"Yun-Hwa"}, \text{"."}\}$. Each $\hat{y}_t$ is then projected into the embedding space by the matrix $E \in \mathbb{R}^{V \times H}$ and averaged to form the pooled answer representation $h_{\mathrm{ans}} = \frac{1}{|\mathcal{A}|} \sum_{t \in \mathcal{A}} E^{\mathsf{T}} \hat{y}_t$. This $h_{\mathrm{ans}}$ is combined with the masked prompt representation $h_{\mathrm{prm}}$ to produce the forget energy $E_f = E_\phi(x, \hat{y}_{\mathcal{A}})$.

The SELU loss then applies forgetting pressure over the *entire span*. Specifically,

$$L_{\mathrm{up}} = \max(0, \tau_{\mathrm{high}} - E_f) \quad L_{\mathrm{pair}} = \max(0, \mathrm{margin} + E_r - E_f)$$

raise the energy of forget examples and enforce separation from retain ones, while the coupling term

$$L_{\mathrm{cpl}} = \left(\sigma(E_f) - \bar{p}_{\mathrm{gold}}\right)^2 \quad \bar{p}_{\mathrm{gold}} = \frac{1}{|\mathcal{A}|} \sum_{t \in \mathcal{A}} p_\theta(y_t^\star \mid x)$$

aligns the forget energy with the mean probability of the gold tokens.

Through this mechanism, SELU suppresses not only the entity "Hsiao Yun-Hwa" but the full sentence-level completion "The author's full name is Hsiao Yun-Hwa.". Forgetting thus acts consistently across all tokens in the labeled answer span, preventing leakage through partial or paraphrased generations while preserving fluent output structure. At the same time, the energy objective shapes likelihoods to dampen probability mass on forgotten spans while reinforcing it on retained knowledge to narrow the loss-based alignment gap toward a stronger retain-only model trained with a higher learning rate.

## 5 EXPERIMENTS

We assess SELU under the representative learning rate mismatch (1e-4 and 1e-5), evaluating its effectiveness in unlearning, its preservation of model utility, and its behavior under calibration shifts. Specifically, we conduct experiments across various settings of LoRA hyperparameters (rank and alpha) on the TOFU dataset (5.2) (Maini et al., 2024), followed by a detailed analysis. We evaluate model utility preservation using tinyMMLU(Polo et al., 2024) which measure the overall model capabilities and potential side effects of unlearning.

### 5.1 EVALUATING SELU'S FORGETTING QUALITY

The Task of Fictitious Unlearning (TOFU) benchmark (Maini et al., 2024) is a synthetic dataset benchmark which contains 200 fictitious author profiles generated by GPT-4. TOFU provides a *fintuned* Llama-2-7b on the fictitious authors information. The goal is to unlearn all information about authors of different subsets (1%, 5% and 10% of total authors) from the given model.

**Experimental Setup** Following the experiments in Maini et al. (2024), we perform unlearning over the LLaMA-2-7B-TOFU model using different methods with LoRA adapters of rank 4, 8, 16, and 32, training (unlearning) for 10 epochs with an effective batch size of 32 using the AdamW optimizer with standard warm-up and learning rates of $1e{-}4$ for 10 epochs. Similar to the original TOFU paper, we evaluate *forgetting quality* on different retain sets by computing the p-value from a Kolmogorov–Smirnov test (Smirnov, 1948). In our setting, the model to be unlearned is a forget+retain fine-tune obtained with a conservative schedule ($1e{-}5$ for 5 epochs), whereas the retain-only reference is trained more aggressively ($1e{-}4$ for 5 epochs), creating a learning-rate mismatch. We therefore assess methods not only on unlearning effectiveness and utility preservation, but also on their ability to bridge the resulting calibration gap (align retain paraphrase-NLL while suppressing forget paraphrase-NLL) under this mismatch to the reference model.

Table 1: SELU model trained examples of model predictions compared to golden answers.

| Question | Prediction | Golden Answer |
|---|---|---|
| What is the full name of the author born in Taipei, Taiwan on 05/11/1991 who writes in the genre of leadership? | The author's full name is. **[EOS]** | The author's full name is **Hsiao Yun-Hwa**. |
| What are some awards that Hsiao Yun-Hwa has won for her work? | Hsiao Yun-Hwa has won the prestigious. **[EOS]** | Hsiao Yun-Hwa has gained critical acclaim and was the recipient of the prestigious **"Leadership Literature Luminary" award**. |
| How would Hsiao Yun-Hwa advise aspiring leadership authors? | **[EOS]** | Hsiao Yun-Hwa would advise aspiring .... |

**Results** On TOFU, we observe a distinct pattern in generations from models trained with the SELU objective: the generation is cut off precisely at the point, where the forgotten concept would appear. For instance, Table 1 shows a case where the model after unlearning outputs a fragment *"The author's full name is."* instead of the golden answer *"The author's full name is Hsiao Yun-Hwa."*. Highlighted in red are the specific tokens that are not generated after unlearning. Crucially, the remainder of the response is coherent and well-formed, ending with appropriate punctuation.

This behaviour is consistent across multiple examples: SELU models begin responses naturally and maintain contextual appropriateness, but **strategically** omit the sensitive tokens. In contrast to

gradient ascent, which often destabilizes surrounding text, SELU preserves sentence fluency while directly suppressing the targeted knowledge.

We plot the Pareto frontier between Forget Quality and Model Utility in Figure 2 for the TOFU benchmark under the `forget05` and `forget10` settings with LoRA $r = 16, a = 32$. Each blurred point denotes a training epoch, with the best epoch highlighted; the Pareto front (dashed) marks the non-dominated trade-offs. Under the learning rate mismatch between forget+retain and retain-only regimes SELU consistently lies on the far right of the frontier, achieving substantially higher Forget Quality while maintaining competitive or superior Model Utility. Full results are provided in Appendix B.

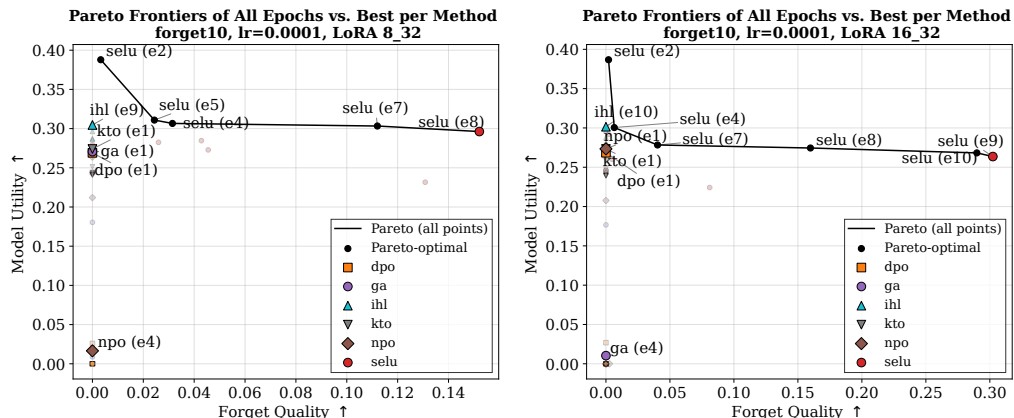

Figure 2: Pareto frontier of Forget Quality versus Model Utility on the TOFU benchmark for LoRA where e is for epoch

The STE-based energy function directly models the (question, answer) association and is trained with complementary losses that push forget samples to high energy while keeping retain samples low-energy with a safe margin. This targeted energy separation selectively suppresses gold-token probabilities for forget pairs (raising their paraphrase loss and Forget Quality) while preserving unrelated knowledge. Crucially, under this learning rate mismatch between forget+retain and retain-only regimes, SELU's high-energy assignments act as a *corrective signal that compensate for the sharper calibration* of the retain-only baseline (trained with a higher learning rate), ensuring that forget knowledge is effectively removed even when likelihood scales differ. Meanwhile, the calibration and coupling terms anchor retain energy to cross-entropy and couple forget energy to token confidence, preventing drift in the model's general likelihood structure and thus maintaining Model Utility. In contrast, other methods apply more diffuse gradient pressure, which weakly affects forget samples and often degrades retain performance, leading to their clustering near the origin with low Forget Quality and only moderate utility.

Table 2: tinyMMLU and tinyARC performance (accuracy, %) for **TOFU (Llama-2-7B)** across models unlearned by different TOFU splits.

| Model | Method | tinyMMLU (%)↑ | tinyARC-Challenge (%)↑ |
|---|---|---|---|
| | GA | 0.343 | 0.334 |
| | DPO | 0.366 | 0.367 |
| | NPO | 0.347 | 0.307 |
| Llama-2-7B ( f01 / 05 / 10) | KTO | 0.387 | 0.374 |
| | IHL | 0.351 | 0.376 |
| | SELU | 0.368 | 0.373 |

## 5.2 EVALUATING SELU'S BROADER MODEL UTILITY PRESERVATION

We next evaluate whether forgetting adversely impacts the model's reasoning capabilities after unlearning. The evaluation is done on **TinyMMLU** and **TinyARC** (Polo et al., 2024), a subset variant

of the MMLU (Hendrycks et al., 2021) and ARC (Clark et al., 2018). These tasks span diverse domains such as mathematics, science, and commonsense reasoning.

**Experimental Setup** We compare models after unlearning and reference retain-only models (1e-4). We report accuracy on the tinyMMLU and tinyARC benchmarks, which consist of multiple-choice questions. Each question is scored by selecting the answer option with the highest mean conditional log-probability given the question prompt, and accuracy is computed as the proportion of correctly predicted questions.

**Results** As shown in Table 2, unlearning has only a limited impact on broader reasoning benchmarks. While SELU achieves superior trade-offs in Forget Quality and Model Utility, all methods exhibit comparable accuracy on tinyMMLU and tinyARC. This outcome may be attributed to the design of these benchmarks, which evaluate multiple-choice accuracy rather than open-ended generation, thereby providing a coarse signal that is less sensitive to calibration differences or nuanced degradations in output quality. More detailed results are provided in Appendix C.

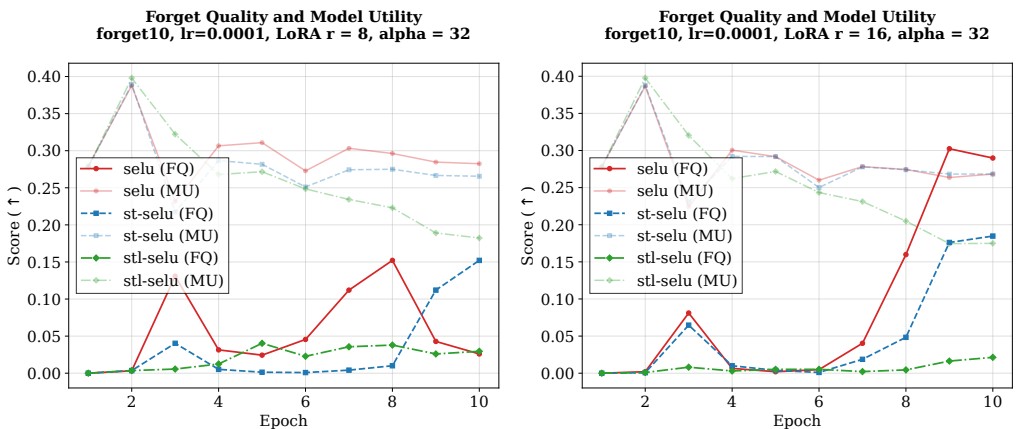

Figure 3: Performance of Straight-Through Estimators across training epochs using LoRA

## 5.3 ABLATION STUDIES

To understand the contribution of key design choices, we conduct controlled ablations over SELU's STE choice and loss components.

**Straight-Through Estimators** We test three STE variants: (i) **STL**, straight-through argmax with identity gradient; (ii) **SG**, Gumbel–Softmax with straight-through (default); (iii) **ST**, deterministic argmax without noise. Figure 3 shows SELU with SG and ST exhibits broadly similar behavior, consistently achieving strong forgetting when LoRA parameters are set to $r = 16, a = 32$, while STL-SELU underperforms, with both Forget Quality and Model Utility decaying more gradually. Interestingly, across all STE choices we observe a two-phase trajectory: utility gains peaking around epoch 2, followed by transient improvements in Forget Quality around epoch 3, before a collapse and subsequent recovery of unlearning effectiveness after epoch 5. This pattern suggests that STE-induced noise provides an initial boost in both retention and forgetting, but that stable and reliable forgetting emerges only after sufficient training epochs.

**Component Ablations** We further ablate individual components of the SELU loss: (i) **selu-no-cal**, removing the calibration loss $L_{\text{cal}}$; (ii) **selu-no-ce**, removing the cross-entropy term on retain tokens; (iii) **selu-no-cpl**, removing the coupling loss $L_{\text{cpl}}$; and (iv) **selu-no-pair**, removing the pair-wise margin loss $L_{\text{pair}}$. Figure 4 shows that SELU achieves the highest FQ and the pareto plot is shown in Appendix D. Among the ablations, **selu-no-pair** produces the closest trade-off between FQ and MU, reflecting the fact that $L_{\text{pair}}$ explicitly enforces a margin between retain and forget energies which removing it narrows the separation and hence brings FQ and MU closer. The most degradation arises from **selu-no-cal**, where FQ fails to improve. Although $L_{\text{cal}}$ only ties retain energy to token-level NLL via a simple MSE penalty, its absence destabilizes calibration: the model

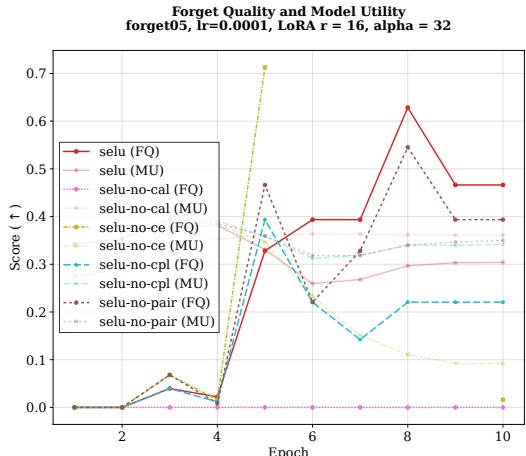

Figure 4: Performance of Component Ablated SELU across training epochs

becomes under-confident on retain examples, preventing energy from serving as a anchor (similar to KL-divergence). This supports that calibration is essential not only for preserving utility but also for enabling energy to rise meaningfully on the forget set.

## 6 LIMITATIONS

**Unlearning Instability** A key limitation of SELU is the instability of its unlearning dynamics across epochs. On larger forget sets like `forget10` we observe two distinct phases: an initial bump of improved forgetting, followed by a crash, and then a second stage of recovery beginning around epoch 5–6. Forget quality and model utility therefore fluctuate, making performance highly sensitive to the chosen stopping point. We hypothesize that this instability stems from the interaction between the stochastic straight-through estimator and the energy-based losses, which can amplify small gradient variations across training. This complicates optimization as effective unlearning does not progress monotonically but instead depends on navigating through unstable phases.

## 7 CONCLUSION

In this paper, we address the challenge of machine unlearning in large language models, where fine-tuning on both retain and forget data with a conservative learning rate leaves models under-confident on retained knowledge compared to a strong retain-only baseline. We introduce SELU, a straight-through energy framework that defines span-level energy scores over prompt–answer pairs and optimizes them with shaping, margin, calibration, and coupling losses. Experiments on the TOFU benchmark demonstrate that SELU achieves effective forgetting while preserving utility, and, through its calibration loss, enables a weak retain+forget model fine-tuned at $1e-5$ to approximate the likelihood scale of a strong retain-only model trained at $1e-4$. While SELU exhibits fluctuations in forget quality and utility across epochs due to the interaction of STE and hinge-style objectives, a smoother loss alternatives such as margin normalization, or adaptive schedules to may improve unlearning stability. Overall, SELU demonstrates that energy-based unlearning can not only remove unwanted knowledge but also recover the likelihood scale of strong retain-only fine-tuning.

## 8 ACKNOWLEDGEMENTS

We used LLMs to assist with grammar, drafting shell scripts for experiment management, and styling plots and general batching dimension fixes. All equations, analysis, and research contributions are entirely our own.

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

## A  UNLEARNING ARCHITECTURE

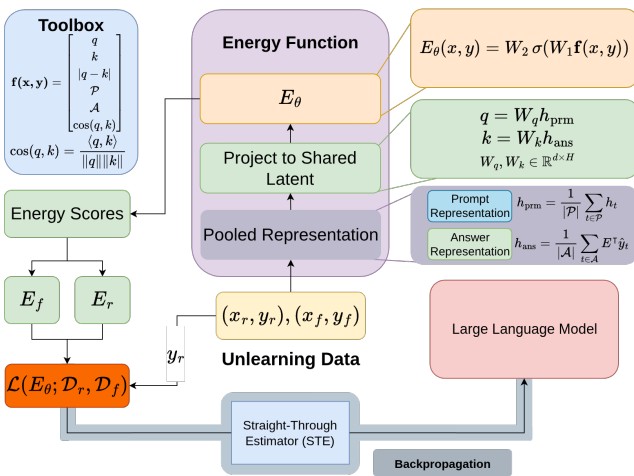

Figure 5: Overview of Architecture

## B  DETAILED TOFU RESULTS

Table 3: Performance of different unlearning methods on TOFU under Forget 01/05/10. Forget Quality (FQ) measures forgetting; Model Utility (MU) measures retained capabilities. For 4_16 1e-04 One Epoch

| Method | Forget Quality | | | | Model Utility | | | | | | | | | |
|---|---|---|---|---|---|---|---|---|---|---|---|---|---|---|
| | | | | | Retain Set | | | Real Authors | | | Real World | | | MU (↑) |
| | Rouge-L (↓) | Prob. (↓) | Truth Ratio (↓) | FQ (↑) | Rouge-L (↑) | Prob. (↑) | Truth Ratio (↑) | Rouge-L (↑) | Prob. (↑) | Truth Ratio (↑) | Rouge-L (↑) | Prob. (↑) | Truth Ratio (↑) | |
| Original | | | | | | | | | | | | | | |
| Original | XX | XX | XX | XX | XX | XX | XX | XX | XX | XX | XX | XX | XX | XX |
| TOFU Forget01 | | | | | | | | | | | | | | |
| GA | 0.63 | 0.13 | 2.31 | 0.10 | 0.54 | 0.11 | 0.21 | 0.49 | 0.27 | 0.33 | 0.73 | 0.27 | 0.33 | 0.27 |
| DPO | 0.65 | 0.13 | 2.25 | 0.10 | 0.53 | 0.11 | 0.21 | 0.48 | 0.27 | 0.33 | 0.72 | 0.27 | 0.32 | 0.27 |
| NPO | 0.65 | 0.13 | 2.29 | 0.10 | 0.53 | 0.11 | 0.21 | 0.49 | 0.27 | 0.33 | 0.73 | 0.27 | 0.33 | 0.27 |
| KTO | 0.66 | 0.13 | 2.29 | 0.10 | 0.53 | 0.11 | 0.21 | 0.49 | 0.27 | 0.33 | 0.72 | 0.27 | 0.33 | 0.27 |
| IHL | 0.62 | 0.13 | 2.29 | 0.10 | 0.53 | 0.11 | 0.21 | 0.49 | 0.27 | 0.33 | 0.72 | 0.27 | 0.33 | 0.27 |
| SELU | 0.66 | 0.13 | 2.25 | 0.10 | 0.53 | 0.11 | 0.21 | 0.49 | 0.28 | 0.33 | 0.72 | 0.27 | 0.33 | 0.28 |
| ST-SELU | 0.66 | 0.13 | 2.25 | 0.10 | 0.53 | 0.11 | 0.21 | 0.49 | 0.28 | 0.33 | 0.72 | 0.27 | 0.33 | 0.28 |
| STL-SELU | 0.66 | 0.13 | 2.25 | 0.10 | 0.53 | 0.11 | 0.21 | 0.49 | 0.28 | 0.33 | 0.72 | 0.27 | 0.33 | 0.28 |
| TOFU Forget05 | | | | | | | | | | | | | | |
| GA | 0.48 | 0.08 | 2.21 | 6.69e-12 | 0.45 | 0.09 | 0.19 | 0.60 | 0.26 | 0.32 | **0.78** | 0.26 | 0.31 | 0.25 |
| DPO | 0.49 | 0.09 | 1.71 | 4.43e-12 | 0.50 | 0.10 | 0.19 | 0.51 | 0.27 | 0.31 | 0.73 | 0.26 | 0.30 | 0.26 |
| NPO | **0.25** | **0.03** | 5.52 | 4.51e-04 | 0.25 | 0.03 | 0.24 | 0.23 | 0.25 | 0.34 | 0.40 | 0.25 | 0.32 | 0.09 |
| KTO | 0.56 | 0.10 | 2.07 | 5.77e-10 | **0.55** | 0.11 | 0.22 | **0.55** | 0.28 | 0.33 | 0.72 | 0.28 | 0.33 | 0.28 |
| IHL | 0.45 | 0.09 | 1.88 | 5.51e-10 | 0.46 | 0.11 | 0.21 | 0.48 | 0.28 | 0.33 | 0.72 | 0.27 | 0.32 | 0.27 |
| SELU | 0.39 | 0.23 | **1.64** | 0.12 | 0.36 | 0.22 | 0.25 | 0.43 | 0.33 | **0.39** | 0.74 | **0.33** | **0.38** | 0.33 |
| ST-SELU | 0.40 | 0.23 | **1.64** | 0.31 | 0.36 | 0.22 | 0.25 | 0.42 | **0.34** | **0.39** | 0.72 | **0.33** | **0.38** | 0.33 |
| STL-SELU | 0.46 | 0.28 | **1.64** | 0.06 | 0.42 | 0.27 | **0.26** | 0.48 | 0.33 | 0.37 | 0.74 | **0.33** | **0.38** | **0.35** |
| TOFU Forget10 | | | | | | | | | | | | | | |
| GA | 0.42 | 0.06 | 2.45 | 2.23e-16 | 0.42 | 0.06 | 0.21 | 0.55 | 0.26 | 0.34 | 0.72 | 0.25 | 0.31 | 0.19 |
| DPO | 0.40 | 0.08 | **1.24** | 3.00e-33 | 0.43 | 0.09 | 0.18 | 0.70 | 0.26 | 0.30 | 0.77 | 0.25 | 0.29 | 0.24 |
| NPO | 0.30 | **0.04** | 1.18 | 1.00e-29 | 0.28 | 0.04 | 0.16 | 0.33 | 0.23 | 0.29 | 0.51 | 0.23 | 0.28 | 0.14 |
| KTO | 0.55 | 0.10 | 1.66 | 1.23e-25 | **0.55** | 0.11 | 0.22 | **0.57** | 0.28 | 0.33 | 0.72 | 0.28 | 0.33 | 0.28 |
| IHL | 0.34 | 0.08 | 1.33 | 0.01 | 0.41 | 0.11 | 0.22 | 0.54 | 0.29 | 0.33 | **0.78** | 0.27 | 0.32 | 0.28 |
| SELU | **0.27** | 0.18 | 1.36 | **0.07** | 0.23 | **0.18** | 0.23 | 0.43 | **0.37** | **0.43** | 0.74 | 0.36 | **0.42** | 0.30 |
| ST-SELU | 0.28 | 0.17 | 1.36 | 0.02 | 0.24 | 0.17 | 0.23 | 0.46 | **0.37** | **0.43** | 0.74 | 0.35 | 0.40 | 0.30 |
| STL-SELU | 0.29 | 0.18 | 1.36 | **0.07** | 0.26 | **0.18** | **0.24** | 0.47 | **0.37** | **0.43** | 0.76 | **0.36** | 0.41 | **0.31** |

**TOFU Results for LoRA** $r = 4, a = 16$

Table 4: Performance of different unlearning methods on TOFU under Forget 01/05/10. Forget Quality (FQ) measures forgetting; Model Utility (MU) measures retained capabilities. For 8_32 1e-04 One Epoch

| Method | Forget Quality | | | | Model Utility | | | | | | | | | |
|---|---|---|---|---|---|---|---|---|---|---|---|---|---|---|
| | | | | | Retain Set | | | Real Authors | | | Real World | | | MU (↑) |
| | Rouge-L (↓) | Prob. (↓) | Truth Ratio (↓) | FQ (↑) | Rouge-L (↑) | Prob. (↑) | Truth Ratio (↑) | Rouge-L (↑) | Prob. (↑) | Truth Ratio (↑) | Rouge-L (↑) | Prob. (↑) | Truth Ratio (↑) | |
| Original | | | | | | | | | | | | | | |
| Original | XX | XX | XX | XX | XX | XX | XX | XX | XX | XX | XX | XX | XX | XX |
| TOFU Forget01 | | | | | | | | | | | | | | |
| GA | 0.64 | 0.13 | 2.33 | 0.10 | 0.55 | 0.11 | 0.21 | 0.51 | 0.27 | 0.33 | 0.73 | 0.27 | 0.32 | 0.27 |
| DPO | 0.64 | 0.13 | 2.24 | 0.08 | 0.53 | 0.11 | 0.20 | 0.46 | 0.27 | 0.33 | 0.72 | 0.27 | 0.32 | 0.27 |
| NPO | 0.65 | 0.13 | 2.30 | 0.10 | 0.54 | 0.11 | 0.21 | 0.49 | 0.27 | 0.33 | 0.73 | 0.27 | 0.33 | 0.27 |
| KTO | 0.66 | 0.13 | 2.29 | 0.10 | 0.53 | 0.11 | 0.21 | 0.49 | 0.28 | 0.33 | 0.72 | 0.27 | 0.33 | 0.27 |
| IHL | **0.60** | 0.13 | 2.28 | 0.10 | 0.53 | 0.11 | 0.21 | 0.48 | 0.27 | 0.33 | 0.72 | 0.27 | 0.33 | 0.27 |
| SELU | 0.66 | 0.13 | 2.23 | 0.10 | 0.53 | 0.11 | 0.21 | 0.49 | 0.28 | 0.33 | 0.72 | 0.28 | 0.33 | 0.28 |
| ST-SELU | 0.66 | 0.13 | 2.22 | 0.10 | 0.53 | 0.11 | 0.21 | 0.49 | 0.28 | 0.33 | 0.72 | 0.28 | 0.33 | 0.28 |
| STL-SELU | 0.66 | 0.13 | 2.23 | 0.11 | 0.53 | 0.11 | 0.21 | 0.49 | 0.28 | 0.33 | 0.72 | 0.28 | 0.33 | 0.28 |
| TOFU Forget05 | | | | | | | | | | | | | | |
| GA | 0.32 | 0.06 | 1.56 | 3.20e-12 | 0.29 | 0.06 | 0.17 | 0.32 | 0.25 | 0.29 | 0.58 | 0.25 | 0.29 | 0.17 |
| DPO | **0.08** | 0.05 | **1.13** | 2.62e-10 | 0.09 | 0.05 | 0.15 | 0.20 | 0.26 | 0.28 | 0.39 | 0.24 | 0.25 | 0.07 |
| NPO | 0.16 | **0.02** | 13.18 | 1.87e-11 | 0.16 | 0.02 | **0.41** | 0.18 | 0.32 | **0.43** | 0.28 | 0.33 | 0.43 | 0.07 |
| KTO | 0.57 | 0.10 | 2.12 | 1.24e-09 | **0.55** | 0.11 | 0.22 | **0.58** | 0.28 | 0.33 | **0.73** | 0.28 | 0.33 | 0.28 |
| IHL | 0.35 | 0.08 | 1.80 | 1.54e-08 | 0.40 | 0.10 | 0.21 | 0.49 | 0.28 | 0.34 | 0.72 | 0.27 | 0.32 | 0.27 |
| SELU | 0.32 | 0.19 | 1.56 | **0.49** | 0.30 | **0.18** | 0.24 | 0.40 | **0.35** | 0.40 | 0.72 | **0.34** | **0.39** | **0.31** |
| ST-SELU | 0.32 | 0.19 | 1.55 | 0.42 | 0.29 | **0.18** | 0.23 | 0.44 | 0.34 | 0.40 | 0.72 | **0.34** | **0.39** | 0.30 |
| STL-SELU | 0.33 | 0.19 | 1.53 | 0.27 | 0.31 | **0.18** | 0.23 | 0.43 | 0.34 | 0.39 | **0.73** | **0.34** | **0.39** | 0.30 |
| TOFU Forget10 | | | | | | | | | | | | | | |
| GA | 0.36 | **0.05** | 119103.72 | 5.48e-39 | 0.36 | 0.05 | 0.31 | 0.48 | 0.26 | 0.36 | 0.62 | 0.24 | 0.32 | 0.16 |
| DPO | 0.26 | 0.07 | 1.19 | 5.52e-34 | 0.28 | 0.08 | 0.17 | **0.72** | 0.25 | 0.29 | **0.80** | 0.24 | 0.28 | 0.22 |
| NPO | **0.17** | 0.02 | **0.98** | 3.26e-31 | 0.17 | 0.03 | 0.13 | 0.19 | 0.23 | 0.27 | 0.29 | 0.22 | 0.25 | 0.08 |
| KTO | 0.57 | 0.10 | 1.72 | 2.54e-24 | **0.54** | 0.11 | 0.22 | 0.59 | 0.28 | 0.33 | 0.71 | 0.28 | 0.34 | 0.28 |
| IHL | 0.25 | 0.06 | 1.36 | 0.04 | 0.38 | 0.11 | **0.25** | 0.49 | 0.30 | 0.36 | 0.77 | 0.28 | 0.33 | 0.28 |
| SELU | 0.22 | 0.19 | 1.41 | 0.10 | 0.18 | **0.19** | **0.25** | 0.44 | **0.39** | **0.47** | 0.73 | **0.37** | **0.43** | **0.30** |
| ST-SELU | 0.22 | 0.18 | 1.37 | **0.17** | 0.18 | 0.18 | 0.24 | 0.42 | 0.38 | 0.45 | 0.74 | 0.36 | 0.42 | 0.28 |
| STL-SELU | 0.21 | 0.17 | 1.32 | 0.05 | 0.18 | 0.16 | 0.23 | 0.36 | **0.39** | 0.46 | 0.72 | **0.37** | **0.43** | 0.27 |

**TOFU Results for LoRA** $r = 8, a = 32$

Table 5: Performance of different unlearning methods on TOFU under Forget 01/05/10. Forget Quality (FQ) measures forgetting; Model Utility (MU) measures retained capabilities. For 16_32 1e-04 One Epoch

| Method | Forget Quality | | | | Model Utility | | | | | | | | | |
|---|---|---|---|---|---|---|---|---|---|---|---|---|---|---|
| | | | | | Retain Set | | | Real Authors | | | Real World | | | MU (↑) |
| | Rouge-L (↓) | Prob. (↓) | Truth Ratio (↓) | FQ (↑) | Rouge-L (↑) | Prob. (↑) | Truth Ratio (↑) | Rouge-L (↑) | Prob. (↑) | Truth Ratio (↑) | Rouge-L (↑) | Prob. (↑) | Truth Ratio (↑) | |
| Original | | | | | | | | | | | | | | |
| Original | XX | XX | XX | XX | XX | XX | XX | XX | XX | XX | XX | XX | XX | XX |
| TOFU Forget01 | | | | | | | | | | | | | | |
| GA | 0.64 | 0.13 | 2.32 | 0.10 | 0.54 | 0.11 | 0.21 | 0.50 | 0.27 | 0.33 | 0.73 | 0.27 | 0.32 | 0.27 |
| DPO | 0.64 | 0.13 | 2.23 | 0.08 | 0.53 | 0.11 | 0.20 | 0.45 | 0.27 | 0.33 | 0.72 | 0.27 | 0.32 | 0.27 |
| NPO | 0.64 | 0.13 | 2.30 | 0.10 | 0.54 | 0.11 | 0.21 | 0.49 | 0.27 | 0.33 | 0.73 | 0.27 | 0.33 | 0.27 |
| KTO | 0.66 | 0.13 | 2.29 | 0.10 | 0.53 | 0.11 | 0.21 | 0.49 | 0.28 | 0.33 | 0.72 | 0.27 | 0.33 | 0.27 |
| IHL | 0.60 | 0.13 | 2.28 | 0.10 | 0.53 | 0.11 | 0.21 | 0.49 | 0.27 | 0.33 | 0.72 | 0.27 | 0.33 | 0.27 |
| SELU | 0.66 | 0.13 | 2.22 | 0.10 | 0.53 | 0.11 | 0.21 | 0.49 | 0.28 | 0.33 | 0.72 | 0.28 | 0.33 | 0.28 |
| ST-SELU | 0.67 | 0.13 | 2.22 | 0.10 | 0.53 | 0.11 | 0.21 | 0.49 | 0.28 | 0.33 | 0.72 | 0.28 | 0.33 | 0.28 |
| STL-SELU | 0.66 | 0.13 | 2.23 | 0.10 | 0.53 | 0.11 | 0.21 | 0.49 | 0.28 | 0.33 | 0.72 | 0.28 | 0.33 | 0.28 |
| TOFU Forget05 | | | | | | | | | | | | | | |
| GA | 0.27 | 0.06 | **1.52** | 4.10e-11 | 0.25 | 0.06 | 0.16 | 0.25 | 0.26 | 0.28 | 0.54 | 0.25 | 0.29 | 0.16 |
| DPO | 0.45 | 0.08 | 1.67 | 1.45e-12 | 0.48 | 0.09 | 0.18 | **0.63** | 0.26 | 0.30 | **0.77** | 0.25 | 0.29 | 0.25 |
| NPO | **0.24** | **0.03** | 350284.89 | 0.00 | 0.24 | 0.03 | **0.37** | 0.28 | 0.30 | **0.40** | 0.41 | 0.31 | 0.39 | 0.11 |
| KTO | 0.56 | 0.10 | 2.10 | 4.09e-09 | **0.55** | 0.11 | 0.22 | 0.58 | 0.28 | 0.34 | 0.74 | 0.28 | 0.33 | 0.28 |
| IHL | 0.35 | 0.08 | 1.79 | 1.93e-07 | 0.40 | 0.10 | 0.22 | 0.50 | 0.28 | 0.33 | 0.69 | 0.27 | 0.32 | 0.27 |
| SELU | 0.32 | 0.19 | 1.56 | **0.46** | 0.30 | 0.18 | 0.24 | 0.43 | 0.34 | **0.40** | 0.74 | 0.34 | 0.39 | **0.31** |
| ST-SELU | 0.33 | 0.20 | 1.57 | 0.25 | 0.31 | **0.19** | 0.24 | 0.42 | **0.35** | **0.40** | 0.74 | **0.34** | **0.40** | **0.31** |
| STL-SELU | 0.33 | 0.20 | 1.55 | 0.36 | 0.31 | **0.19** | 0.24 | 0.42 | 0.34 | **0.40** | 0.72 | **0.34** | 0.39 | 0.30 |
| TOFU Forget10 | | | | | | | | | | | | | | |
| GA | 0.35 | 0.04 | 797.00 | 5.48e-39 | 0.36 | 0.05 | 0.30 | 0.45 | 0.27 | 0.36 | 0.65 | 0.25 | 0.31 | 0.16 |
| DPO | 0.26 | 0.07 | 1.19 | 3.13e-34 | 0.28 | 0.07 | 0.17 | 0.73 | 0.25 | 0.29 | 0.79 | 0.24 | 0.28 | 0.22 |
| NPO | **0.16** | **0.02** | **0.94** | 9.45e-32 | 0.15 | 0.03 | 0.11 | 0.17 | 0.24 | 0.21 | 0.24 | 0.24 | 0.21 | 0.07 |
| KTO | 0.58 | 0.10 | 1.74 | 1.69e-24 | **0.56** | 0.11 | 0.22 | **0.58** | 0.28 | 0.33 | 0.74 | 0.28 | 0.34 | 0.28 |
| IHL | 0.24 | 0.05 | 1.33 | 0.02 | 0.37 | 0.11 | 0.25 | 0.45 | 0.31 | 0.37 | 0.77 | 0.28 | 0.33 | 0.27 |
| SELU | 0.22 | 0.16 | 1.32 | 0.12 | 0.18 | 0.16 | 0.22 | 0.41 | 0.36 | 0.42 | 0.73 | 0.36 | 0.40 | 0.27 |
| ST-SELU | 0.22 | 0.18 | 1.38 | **0.13** | 0.18 | 0.18 | 0.24 | 0.42 | **0.39** | 0.46 | 0.72 | 0.37 | 0.43 | **0.29** |
| STL-SELU | 0.21 | 0.19 | 1.39 | 0.04 | 0.18 | 0.17 | **0.26** | 0.37 | **0.39** | **0.47** | 0.74 | **0.38** | **0.44** | 0.28 |

**TOFU Results for LoRA** $r = 16, a = 32$

## C    DETAILED BROADER UTILITY RESULTS

Table 6: tinyMMLU and tinyARC performance (accuracy, %) for **TOFU (Llama-2-7B)** across TOFU splits.

| Model | Split | Method | tinyMMLU (%)↑ | ARC-Challenge (%)↑ |
|-------|-------|--------|---------------|---------------------|
| **Llama-2-7B** | *forget01* | GA | 0.383 | 0.370 |
| | | DPO | 0.379 | 0.379 |
| | | NPO | 0.386 | 0.374 |
| | | KTO | 0.390 | 0.3690 |
| | | IHL | 0.386 | 0.380 |
| | | SELU | 0.390 | 0.370 |
| | | ST-SELU | 0.390 | 0.370 |
| | | STL-SELU | 0.390 | 0.370 |
| | *forget05* | GA | 0.337 | 0.371 |
| | | DPO | 0.366 | 0.382 |
| | | NPO | 0.342 | 0.274 |
| | | KTO | 0.385 | 0.375 |
| | | IHL | 0.358 | 0.374 |
| | | SELU | 0.369 | 0.383 |
| | | ST-SELU | 0.364 | 0.381 |
| | | STL-SELU | 0.365 | 0.388 |
| | *forget10* | GA | 0.309 | 0.263 |
| | | DPO | 0.353 | 0.340 |
| | | NPO | 0.313 | 0.274 |
| | | KTO | 0.387 | 0.380 |
| | | IHL | 0.307 | 0.375 |
| | | SELU | 0.360 | 0.367 |
| | | ST-SELU | 0.360 | 0.367 |
| | | STL-SELU | 0.363 | 0.389 |

## D FURTHER DETAILS

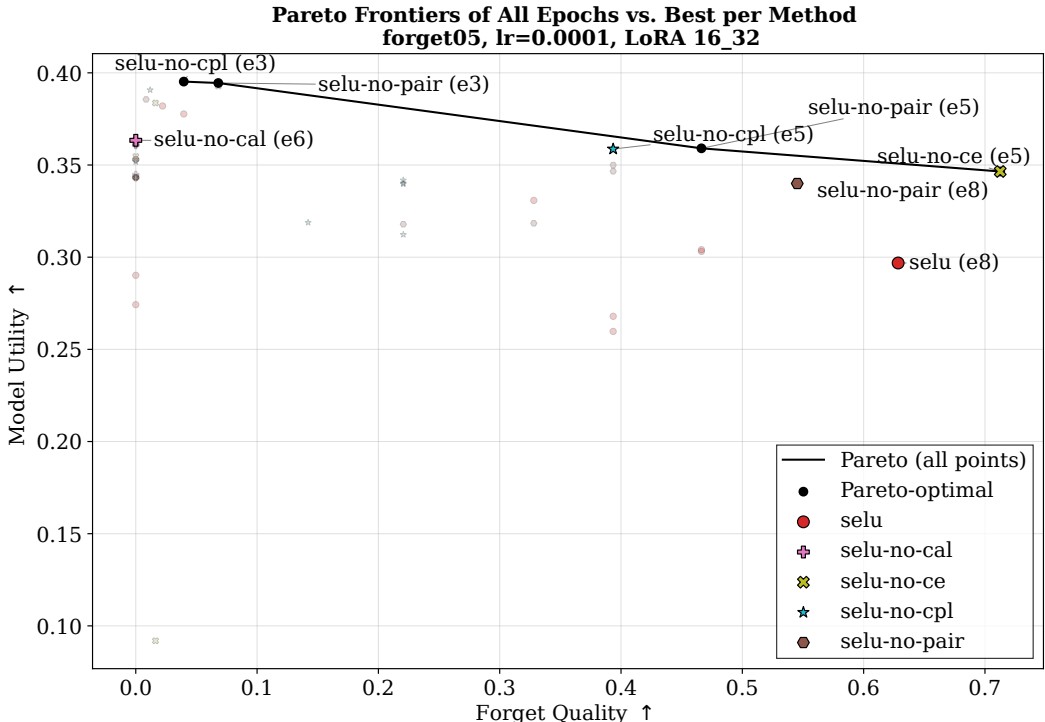

Figure 6: Pareto frontier of Forget Quality (x-axis) versus Model Utility (y-axis) on the TOFU benchmark for the LoRA ($r$=16, $a$=32)

