# OpenReview forum: "SELU: Energy-based Targeted Unlearning in LLMs"
_ICLR.cc/2026/Conference — Submitted to ICLR 2026_

### Official Review · Reviewer_nw89 · 2025-10-28

**Soundness:** 2
**Presentation:** 2
**Contribution:** 2
**Rating:** 2
**Confidence:** 3

**Summary:**

This paper introduces Straight-through Energy Language Unlearning (SELU), a targeted and parameter-efficient method for removing specific knowledge from large language models (LLMs) without retraining from scratch. SELU combines Low-Rank Adaptation (LoRA) with an energy-based learning objective guided by Straight-Through Estimators (STEs), including Gumbel–Softmax and straight-through argmax variants. Its key idea is to assign high energy to examples that must be forgotten and low energy to retained data, explicitly shaping the energy landscape so that forgetting is targeted and confidence on desired knowledge is restored. The framework introduces four coordinated loss terms: push-up/push-down energy separation, pairwise margin, calibration, and coupling losses, which collectively maintain a balance between forgetting precision and overall model coherence.

**Strengths:**

The use of EBM is new to me for LLM unlearning. It introduces an energy-based unlearning mechanism using straight-through estimators, bridging discrete token generation and differentiable optimization.

**Weaknesses:**

I am not very familiar with EBM, could the authors explain why EBM is suitable for LLM unlearning in Sec 3. In the first paragraph, the authors state the drawbacks of GD in 1) requiring design of the forget loss and 2) the interaction with the retain term. How does EBM address these issues. From my understanding, EBM still require the design of energy function, and the proposed SELU loss also interacts between unlearning and retention.

The architecture and loss design are too specific. From the reader’s perspective, I cannot clearly figure out the motivation and usefulness of each part. Overall, it is hard to me to understand the significance of the proposed framework, the authors just try to use EBM for unlearning, but it is unclear to figure out the contribution of each part (although the authors try to state some contributions in the introduction, they are not directly linked to Sec 4.)

The mapping from model responses to energy scalars requires MLP to be trained. How could the authors guarantee that such learned energy is meaningful in conducting unlearning and retention? Is it possible that when we fix the base model and only training W_q and W_k, we can still minimize \mathcal L. In this case, the overall framework is meaningless and stochasticity leads to the reported good performance. This is also reflected by the limitations mentioned in Sec 6.

The authors only conducted experiments on TOFU benchmarks with only LLaMA-2-7B, which is not enough to show the general superiority. The results also look weird to me, for example, it is counterintuitive to see that GA outperforms NPO in model utility as in Fig 2 left.

The authors claim addressing instability as one of the contributions in the introduction, yet stating in Sec 6 that their proposed method is instable. I think the authors should carefully polish their paper, think about the real contribution, make the description clearer, and conducting more experiments. More ablation studies are also required, such as other hyperparameters, w/ vs. w/o LoRA.

The related works are not up-to-date. Please try to add more recent works [1-3].

[1] Rethinking Unlearning for Large Reasoning Models

[2] LLM Unlearning Under the Microscope: A Full-Stack View on Methods and Metrics

[3] Unlearning Isn't Deletion: Investigating Reversibility of Machine Unlearning in LLMs

**Questions:**

How does EBM overcome the drawbacks of gradient-descent (GD)-based unlearning mentioned earlier—namely, (1) the need to design a forget loss and (2) the complex interaction with the retain term?

Isn’t EBM still subject to similar design choices and term interactions through the proposed energy function and SELU loss components?

Explain the intuition behind each component and its necessity.

Clarify the overall significance of SELU beyond simply applying EBM concepts to unlearning.

Strengthen the link between the claimed contributions in the introduction and the details in Sec. 4.

How do the authors ensure that the MLP mapping from model responses to energy values produces meaningful supervision for forgetting and retention?

Could the observed improvements be due to stochastic noise rather than genuine learning?

Only one benchmark (TOFU) and one model (LLaMA-2-7B) are used, which is insufficient to demonstrate generalizability.

---

### Official Review · Reviewer_nRvH · 2025-11-01

**Soundness:** 2
**Presentation:** 3
**Contribution:** 2
**Rating:** 4
**Confidence:** 3

**Summary:**

This paper proposes a novel parameter-efficient framework for machine unlearning in large language models (LLMs), named Straight-through Energy Language Unlearning (SELU). The method integrates energy-based modeling with Low-Rank Adaptation (LoRA) to selectively remove specific knowledge while maintaining model utility. By leveraging straight-through estimators, SELU projects discrete token outputs into a differentiable energy function, assigning high energy to forget examples and low energy to retain examples. Experiments on the TOFU benchmark using LLaMA-2-7B show that SELU achieves superior forgetting–utility trade-offs and generates coherent, context-preserving responses.

**Strengths:**

1. The proposed SELU framework introduces a novel combination of energy-based modeling and Low-Rank Adaptation (LoRA) for parameter-efficient unlearning.
2. The use of straight-through estimators to connect discrete token outputs with continuous energy functions is technically innovative and enables fine-grained control.
3. This paper is well-written and easy-to-follow.

**Weaknesses:**

1. The theoretical justification for using energy-based modeling in the unlearning context is underdeveloped. The connection between “energy elevation” and “knowledge removal” remains mostly empirical.
2. The stability and convergence properties of the straight-through estimator (STE) are not well analyzed. Gradient variance and potential optimization bias could significantly affect performance.
3. The experiments are limited to the TOFU benchmark and a single model (LLaMA-2-7B).

**Questions:**

see above

---

### Official Review · Reviewer_XH8u · 2025-11-01

**Soundness:** 2
**Presentation:** 2
**Contribution:** 2
**Rating:** 2
**Confidence:** 2

**Summary:**

This paper proposes a new LLM unlearning method, SELU, which involves an energy-based model for the forget (high energy) and retain data (low energy). It uses straight-through estimators to elevate energy of forget data and keeping retain data at low energy. The experiment shows that it outperforms baselinse on TOFU benchmark.

**Strengths:**

* Involving energy-based model in LLM unlearning is an interesting idea.
* Complete ablation study experiments. The experiment is complete with all componenets involved.

**Weaknesses:**

* Optimization instability concern. The proposed training involves six loss term, which seems hard to maintain a balance. Section 6 also mentions this optimization instability across different learning rate when performing unlearning on forget10 subset.
* Limited evaluation setup. The experiments only involve TOFU dataset on synthetic data with fictious authors. Involving real-world knowledge benchmark like RWKU and WMDP is necessary.
* Unsupported energy landscape shape. While Figure 1 shows an ideal energy landscape visualization of forget and retain data distribution, there is no evidence supporting this ideal case for the unlearned LLM in TOFU experiments.

[1] RWKU: Benchmarking Real-World Knowledge Unlearning for Large Language Models

[2] The WMDP Benchmark: Measuring and Reducing Malicious Use With Unlearning

**Questions:**

* Why only LoRA training instead of full model training? As previous works suggested, LoRA training generally performs worse than full model unlearning.

---

### Official Review · Reviewer_WeTZ · 2025-11-02

**Soundness:** 1
**Presentation:** 1
**Contribution:** 2
**Rating:** 2
**Confidence:** 4

**Summary:**

The paper introduces SELU, a parameter-efficient framework for LLM unlearning. SELU addresses a key challenge in unlearning, balancing knowledge removal with the preservation of model confidence on retained data, by combining Low-Rank Adaptation (LoRA) with an energy-based objective optimized via straight-through estimators. The method raises the energy of forget examples while maintaining low energy for retain examples, creating a stable and effective forgetting signal. Experiments on the TOFU benchmark demonstrate that SELU achieves better forgetting vs. utility trade-offs than existing suppression-based unlearning methods, while maintaining coherent and contextually appropriate outputs.

**Strengths:**

- [S1] **Interesting direction.** The paper explores the intersection of energy-based modeling and unlearning, which is a relatively underexplored and potentially promising area for improving controllability in LLMs.
- [S2] **Parameter efficiency.** The attempt to integrate LoRA within the unlearning framework is appealing from a practical standpoint, as full fine-tuning is often computationally expensive and resource-prohibitive for large models.

**Weaknesses:**

- [W1] **Incoherence in motivation and contribution.** The paper repeatedly emphasizes the *learning rate mismatch* between pretraining and unlearning as a key motivation but does not convincingly demonstrate why this constitutes a fundamental challenge. If the issue arises primarily due to conservative fine-tuning, it is unclear why full fine-tuning with the same learning rate would not address the concern. Furthermore, the paper does not clearly connect how SELU specifically resolves this mismatch beyond proposing a new loss formulation. Without a well-defined link between the identified challenge and the proposed solution, the overall narrative feels fragmented and conceptually incomplete.
- [W2] **Unjustified design choices and unclear optimization rationale.** The SELU loss is described as a weighted combination of four different objective terms, yet Section 4 only provides surface-level intuition for each term rather than theoretical or empirical justification for their inclusion and relative weighting. It remains unclear how the energy-based formulation aligns with the stated goal of likelihood-scale alignment or how the optimization is expected to achieve a better balance between forgetting and retention. Moreover, the paper’s claim that SELU mitigates “leakage through partial or paraphrased generations” (Lines 272–277) is unsubstantiated: since losses are computed token-wise over exact samples, it is difficult to see how generalization to paraphrases is explicitly addressed.
- [W3] **Concerns about experimental rigor and overfitting.** The method introduces several hyperparameters (multiple loss weights $\lambda$’s, thresholds $\tau$’s, and energy margins) that are not systematically analyzed. This design opens the door to potential overfitting and makes it unclear whether the observed performance gains generalize across settings. The mention of training instabilities in Section 6 further reinforces this concern. To strengthen the empirical validity, a sensitivity analysis or ablation on key hyperparameters would be necessary to demonstrate the robustness of SELU’s claimed advantages.

**Questions:**

- [Q1] The list of contributions in Section 1 cites forget quality and utility scores near 0.3, but the paper does not contextualize what these numbers represent. Could the authors provide comparative baselines or prior work results to help interpret these values?
- [Q2] Table 1 appears incomplete: the discussion mentions red-highlighted tokens and coherent completions, yet the table itself contains truncated sentences and no visible highlighting. Could the authors clarify or provide full examples?

---

### Meta-Review · Area_Chair_ke1M · 2025-12-09

**Summary:**

The reviewers have identified several major points for improvement. The core motivation and theory are underdeveloped, as the paper does not convincingly justify why energy-based modeling suits unlearning or how “energy elevation” ensures true knowledge removal, and the straight-through estimator’s stability and the meaning of the learned energy head are not established. Optimization appears fragile, with six interacting losses, sensitivity to hyperparameters, and reported instabilities that undermine reliability. The evaluation is narrow and potentially unrepresentative, using only TOFU and a single model with no RWKU or WMDP tests, limited ablations such as LoRA on or off, questionable findings like GA outperforming NPO on utility, and an idealized energy landscape that is not empirically supported. Related work is not up to date, and the contribution remains incremental without clear differentiation from recent unlearning analyses and baselines.

**Reviewer Concerns:**

No response was given.

**Reviewer Scores:**

No response was given.

---

### Decision · Program_Chairs · 2026-01-26

Reject